# Video Active Perception: Efficient Inference-Time Long-Form Video Understanding with Vision-Language Models

## Abstract

Large vision-language models (VLMs) have advanced multimodal tasks such as video question answering (QA). However, VLMs face significant challenges with long-form videos due to the prohibitive computational costs of processing extremely long token sequences. Inspired by active perception theory, which posits that models gain information by acquiring data that differ from their expectations, we introduce Video Active Perception (VAP), a training-free method to enhance long-form video QA using VLMs. Our approach treats key frame selection as data acquisition in active perception and leverages a lightweight text-conditioned video generation model to represent prior world knowledge. Empirically, VAP achieves state-of-the-art zero-shot results on long-form video QA datasets such as EgoSchema, NExT-QA, ActivityNet-QA and CLEVRER, achieving an increase of up to $5.6\times$ efficiency by frames per question over standard GPT-4o, Gemini 1.5 Pro, and LLaVA-OV. Moreover, VAP shows stronger reasoning abilities than previous methods and effectively selects key frames relevant to questions. These findings highlight the potential of leveraging active perception to improve efficiency and effectiveness of long-form video QA.

## 1 Introduction

Multimodal foundational models, particularly large vision-language models (VLMs) (Achiam et al., 2023; Reid et al., 2024), have achieved remarkable results in tasks such as image captioning, text-to-image generation, and video question answering. Long-form video question answering (Xiao et al., 2021; Mangalam et al., 2024) stands out as a challenging and intriguing problem. It requires models to reason over complex dynamics, intricate scenes, and subtle visual details across extended time frames, akin to how humans extract information from complicated visual streams. Developing effective solutions for this task is practically important and poses compelling scientific challenges.

However, the large number of tokens generated from long videos poses a significant bottleneck for existing VLM approaches, especially during inference. For example, processing one hour of 720p video can produce nearly four million tokens, and performing inference on a 100-hour video once could cost almost \$2,000 [1]. Real-world applications such as autonomous driving, long-term patient monitoring, or surveillance analysis often involve thousands or millions of hours of video data. These prohibitive inference costs hinder the practical deployment of VLMs in real-world video tasks.

In this paper, we draw inspiration from "active perception" (Bajcsy, 1988; Aloimonos, 2013; Bajcsy et al., 2018), which posits that intelligent agents should actively acquire data that differ from their prior beliefs or models of the world. As Bajcsy et al. (2018) articulates, "an agent is an active perceiver if it knows why it wishes to sense, and then chooses what to perceive, and determines how, when and where to achieve that perception." This concept mirrors the mechanisms of active perception in the human brain (Tenenbaum, 1971; McArthur & Baron, 1983; Dijksterhuis, 2001; Satchell et al., 2021; Wu et al., 2024), where we continuously compare reality with our expectations, identify discrepancies, and seek additional information. The essence of active perception is to *guide data acquisition through a priori knowledge of the world.*

---

[1]Assuming sampled at 1 frames per second; specifically, $3,978,000$ tokens for a one-hour video. Pricing with GPT-4o estimated from `https://openai.com/api/pricing/`.

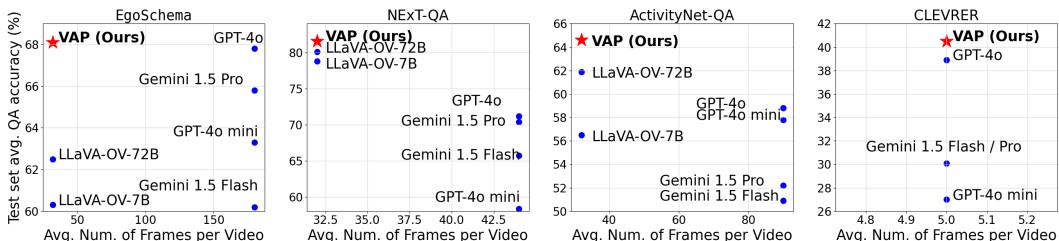

Figure 1: VAP outperforms standard uniform sampling GPT-4o, Gemini 1.5 and LLaVA-OV on EgoSchema, NExT-QA, ActivityNet-QA and CLEVRER with up to 5.6× more efficient on frame per question.

Building on these principles, we introduce Video Active Perception (VAP), a training-free method that improves both efficiency and performance for long-form video question answering. Concretely, the *data acquisition* aspect of active perception corresponds to *selecting key video frames* in videos for VLM inference, while the *a priori knowledge* is embodied by a lightweight, pre-trained text-conditional video generation model that encodes complex prior visual knowledge of the world.

To select frames, VAP begins by sparsely sampling a few initial frames from the video. These frames, along with the question and possible answers, are fed into the generation model as conditional signals to produce unseen video frames in the latent space. Simultaneously, all real frames are efficiently encoded into the latent space, resulting in two sets of latents: generated latents representing expected video dynamics and actual latents representing real scenes and transitions. By comparing these two sets, VAP identifies the real frames that diverge the most from the generated latents, those that are most "surprising" relative to the prior knowledge, and selects them as key frames for VLM inference. VAP prioritizes informative input and enhances efficiency. Unlike previous frame selection methods (Wang et al., 2024a; Fan et al., 2024; Wang et al., 2024b), VAP does not require a captioning model and operates in a single selection round rather than through multi-round selection or complex data structures, providing a simplified, unified approach.

Empirically, by harnessing the principles of active perception, VAP demonstrates substantial improvements in both efficiency and performance across long-form video QA datasets such as EgoSchema (Mangalam et al., 2024), NExT-QA (Xiao et al., 2021), ActivityNet-QA (Yu et al., 2019), and the reasoning dataset CLEVRER (Yi et al., 2019). By selectively focusing on frames that diverge from prior knowledge, VAP outperforms standard uniform sampling methods using GPT-4o, Gemini 1.5 Pro, and LLaVA-OV-72B (Li et al., 2024a), achieving up to a 5.6× increase in efficiency by frames per question (see Figure 1 ). It obtains state-of-the-art zero-shot results with 68.1% on EgoSchema, 81.4% on NExT-QA, 64.6% on ActivityNet-QA, and 40.5% on CLEVRER (Table 1). Notably, VAP is both VLM-agnostic and task-agnostic, making it applicable across various vision-language models and video QA tasks. Our quantitative analysis reveals that VAP exhibits stronger visual reasoning capabilities than previous caption-based models on challenging temporal and causal reasoning tasks (Section 3.5). Qualitative results illustrate that VAP effectively selects unseen pivotal frames relevant to the questions for explanatory and counterfactual reasonings (Section 3.6).

Our contributions are threefold:

- Inspired by the active perception theory, we introduce Video Active Perception (VAP), a method that *enhances the efficiency and performance* of long-form video question answering during VLM inference. By selecting key frames that most significantly diverge from those generated by a lightweight pre-trained video generation model conditioning on questions and answers, VAP focuses on the most informative content.

- VAP *outperforms standard GPT-4o, Gemini 1.5 Pro and LLaVA-OV-72B*, achieving up to a 5.6× improvement in efficiency regarding frames per question on datasets on EgoSchema, NExT-QA, ActivityNet-QA and CLEVRER. It also surpasses recent frame selection baselines for VLM inference, effectively selecting question-relevant frames and outperforming caption-based methods on challenging visual reasoning tasks.

- These findings demonstrate the *efficacy of leveraging prior world knowledge from a generation model* to enhance both the efficiency and effectiveness of long-form video question

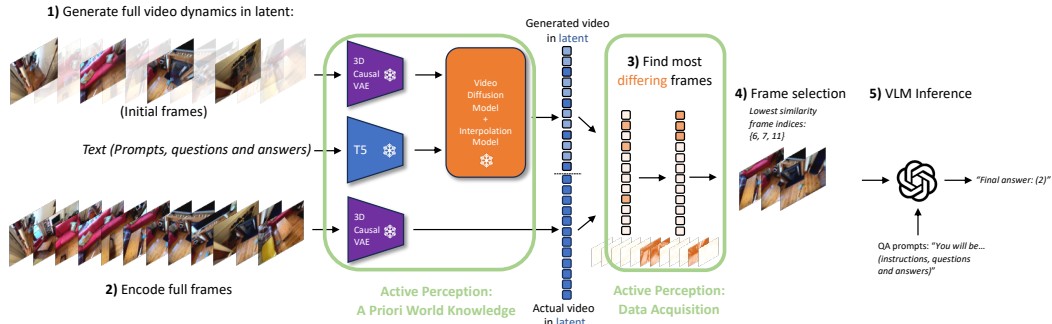

Figure 2: Overview of Video Active Perception model (VAP). There are two core modules in VAP: an a priori knowledge model, and a data acquisition process. The a priori knowledge model, which contains extensive visual knowledge, generates full video dynamics from a few initial frames and QA information. The data acquisition process compares the generated vs. real video dynamics and finds actual frames that are most informative based on difference from the expected video dynamics. The selected frames are used for the VLM inference.

answering, highlighting the potential for more intelligent data acquisition strategies when performing inference on existing large Vision-Language Models.

## 2 VIDEO ACTIVE PERCEPTION

In this section, we introduce the key technical components of Video Active Perception: a) a priori world knowledge: producing the full video dynamics in the latent space via the lightweight text-conditioned generative model, in Section 2.1); b) data acquisition: frame selection based on the comparison between the generated and real frames, in Section 2.2; and c) VLM inference: running inference on flagship VLMs from the selected frames, in Sec 2.3.

We formulate training-free, inference-time usage instead of fine-tuning or pre-training large Vision-Language Models (VLMs) for video question answering. We have a long-form video $x_{1:T} = (x_1, \ldots, x_T)$ with frames $x_i$, and a large total number of frames $T$. The video comes with the question $q$, possible answers $a$, and the user-defined prompt to facilitate VLMs. We assume a lightweight pre-trained text-conditioned video generation model $f(\cdot)$, which takes a sequence of $t$ video frames $x_{1:t}$, where $t$ is the number of frames and the text prompt $p$, and output latents with full frame numbers $\tilde{h}_{1:T}$ (details explained below). Our goal is to select a subset of $K$ frames from $x_{1:T}$ to perform inference efficiently with a large VLM $g(\cdot)$. The overview algorithm of our method is given in Algorithm 1, and we will break it down into the following sections.

### 2.1 A PRIORI KNOWLEDGE FOR GENERATING FULL VIDEO DYNAMICS

We present the a priori knowledge module in VAP: generating video dynamics by producing unseen frames from only a few frames along with the question and answers. VAP uses a pre-trained, lightweight video generation model to encode the seen frames and texts, and a frame interpolation module to produce the latents corresponding to the unseen frames. Then, all latents (seen and generated frames, and text) will be passed to a stack of transformer blocks for better alignment of visual and textual feature spaces. The model output are unpatchified to the original latent shape, and will be fed into a denoiser of a video diffusion model as conditioning signals to sample latent frames. For the generation model, we use CogVideo (Hong et al., 2022) and its updated variation, CogVideoX (Yang et al., 2024), as the video generation model. CogVideo is an open-source large-scale diffusion transformer for general text-to-video generation. CogVideoX (Yang et al., 2024), based on CogVideo, and is a state-of-the-art large-scale diffusion transformer model for text-conditioned video generation. We use CogVideoX for encoding due to better compression from pixels to latent spaces, and the recursive interpolation module from CogVideo for frame interpolation. We list the key steps below, and the training details of CogVideoX can be found in Yang et al. (2024).

**Uniform sampling initial frames.** First, we uniformly sample a small subset of initial frames from the video as bare bones for generation. These frames are supposed to provide the basic dynamics and visual context of the videos. We sample up to $n = 32$ frames in our experiments, a small amount compared to long videos (e.g., $5,400$ frames in one video from EgoSchema).

---

**Algorithm 1** Algorithm of VAP.

---

**Require:** Inference video $\boldsymbol{x}_{1:T}$, prompt $\boldsymbol{p}$, question $\boldsymbol{q}$ and answers $\boldsymbol{a}$, video generation model $f(\cdot)$, and a large Vision-Language Model (VLM) $g(\cdot)$, initial frame number $k$, final frame number $n$.
1: Uniformly sample $k$ initial frames from $\boldsymbol{x}_{1:T}$: $\{\boldsymbol{x}_i\}_{i \in \mathcal{S}}$, where $\mathcal{S} \subseteq \{1, 2, \ldots, T\}, |\mathcal{S}| = k$
2: Get a set of latents by encoding initial frames and then sampling from the generation model: $\tilde{\boldsymbol{h}}_{1:T} \leftarrow f\left(\{\boldsymbol{x}_i\}_{i \in \mathcal{S}} | \boldsymbol{q}, \boldsymbol{a}\right)$
3: Get a set of latents from encoding all real frames: $\boldsymbol{h}_{1:T} \leftarrow f(\boldsymbol{x}_{1:T} | \boldsymbol{q}, \boldsymbol{a})$
4: Compute the cosine similarity $\boldsymbol{c}$: $c_i \leftarrow \boldsymbol{h}_i \cdot \tilde{\boldsymbol{h}}_i$
5: Sort and find $n$ indices with lowest similarities: $\mathcal{I}_n \leftarrow \underset{i_1, \ldots, i_n}{\arg\min} (c_i)$.
6: Select $k$ frames by $\mathcal{I}_n$: $\{\boldsymbol{x}_i | i \in \mathcal{I}_n\}$
7: Return response from VLM with selected frames:
8: **return** $g\left(\boldsymbol{p}, \boldsymbol{q}, \{\boldsymbol{x}_i | i \in \mathcal{I}_n\}\right)$.

---

**Encoding sampled frames.** We adopt the pre-trained 3D VAE (Yu et al., 2023; Yang et al., 2024), which incorporate 3D convolutions to compress video spatially and temporally to achieve higher compression ratio for improved video reconstruction quality and continuity. The encoder of the 3D VAE each contains four $2 \times$ downsampling, with both the spatial and temporal dimension in the first rounds and spatial dimension only in the last round. This achieves a $4 \times 8 \times 8$ compression from pixels to latents. This design is crucial for compressing enough visual information to make a rich, contextualized generation of unseen frames in latent space possible.

**Alignment betweeen vision and text.** The latents of the sampled frames and the interpolated frames are combined by their interpolation ordering, patchified along the spatial dimension. Following Yang et al. (2024), a 3D Rotary Position Embedding (Su et al., 2024), a relative position encoding that is better than the sinusoidal absolute position encoding (Yang et al., 2024), is applied to the spatio and temporal dimensions. The text input is encoded using T5 (Raffel et al., 2020). The latents of both modality are fed into a stack of diffusion transformer blocks. Following DiT (Peebles & Xie, 2023), we also use the timestep $t$ of the diffusion process as the input to the transformer. Modality-specific adaptive layer norms are applied to both the video and the text latent, which is shown to promote the alignment of feature spaces across modalities (Yang et al., 2024). A 3D text-video hybrid attention mechanism is used, as in Yang et al. (2024).

**Generating latent frames.** Next, we seek to sample (generate) video frames from the video diffusion module (Ho et al., 2022) of CogVideoX, but only in latent space. First, the outputs of the last step are unpatchified to restore the original latent shape. Then, we simply sample from the video diffusion model by feeding the aligned visual-textual latents as the conditioning signal. We then leverage the pre-trained frame interpolation model from CogVideo (Hong et al., 2022), which is based on Real-Time Intermediate Flow Estimation (RIFE) (Huang et al., 2022). The frames in latent space are chunked into frame blocks, and for each block, a frame in latent space is interpolated with guidance of structural similarity index measure (SSIM). SSIM threshold can be adjusted to reach the desired number of frames. For very long videos where the number of frames is large, we leverage a memory bank to cache the latents. The RIFE model is fast and the interpolation is light-weighted, and we regard it as an integral part of the generation model.

## 2.2 DATA ACQUISITION BY VIDEO FRAME SELECTION

The next step is data acquisition, which involves selecting the most informative and "surprsing" frames. From active perception principles, these are the frames that diverge the most from the expected video dynamics produced by the generation model.

**Encoding all real frames.** We use the 3D VAE encoder of the same video generation model, CogVideoX, to encode all real frames. Doing so allows us to compare real frames against generated frames in an efficient way, since all encoded frames are in latent space only. The 3D VAE achieves $4 \times 8 \times 8$ compression, and we use a memory bank for long videos from EgoSchema to store the latents. Note that we did not leverage the sampling capabilities of the generation model (i.e., the video diffusion model) to further improve efficiency.

**Similarity between real and generated frames.** Given the real frames in the latent space and the generated frames in the latent space, we now acquire the most informative real frames by comparing

the two sets of latent frames against each other. In this paper, we use cosine similarity to select the most dissimilar real frame from their generated counterparts. All real frames will have one corresponding generated frame in latent space; therefore, we compute the cosine similarity for each pair. We then sort the cosine similarity list and select the indices of $n = 32$ real frames with the lowest similarities to their generated counterparts for VLM inference. In Section 2 we explore different frame numbers and find $n = 32$ sufficiently optimal for our tasks.

## 2.3 VLM INFERENCE FROM SELECTED FRAMES

After locating the frame indices, we combine the frames into a set. With this set and an instructional prompt, the question and all possible answers, we use them to run VLM inference. We include the exact formats of the prompts in the Appendix (Section A.1). All VLMs we use (GPT-4o, GPT-4o mini, GPT 1.5 Pro, GPT 1.5 Flash, LLaVA OneVision-7B and LLaVA-OneVision 72B) can take interleaved prompts of images and texts as input. After inference with VLM on each video QA dataset, we gather the generated responses, parse the answers, and evaluate the results.

## 3 EXPERIMENTS

In this section, we introduce the datasets, evaluation metrics, baselines, implementation details, main results of VAP, and quantitative and qualitative analyzes.

### 3.1 DATASETS AND METRICS

**EgoSchema.** Egoschema (Mangalam et al., 2024) is a dataset with $5,000$ videos, along with an associated question and answer pair. Each question is a multiple-choice question that has a series of five answers that are associated with it. Each video clip is around three-minutes long. The videos in EgoSchema cover a range of topics, including different types of human behavior. In order to evaluate the model's performance on EgoSchema, we measure the percentage of predicted multiple choice answer options that match the correct multiple choice answer option. These multiple choice answer accuracy can also be measured by submitting to the EgoSchema leaderboard (Kaggle, 2024).

**NExT-QA.** NExT-QA (Xiao et al., 2021) is a dataset to study causal and temporal action reasoning in video. It contains $5,440$ videos and about $52,000$ manually annotated question-answer pairs grouped into causal, temporal, and descriptive questions. There are multi-choice QA provides five candidate answers, as well as open-ended QA. This dataset challenges models to truly understand the causal and temporal structure of the actions.

**ActivityNet-QA.** ActivityNet-QA (Yu et al., 2019) consists of 58,000 QA pairs on 5,800 complex web videos derived from the popular ActivityNet (Caba Heilbron et al., 2015) dataset, which contains diverse web videos with two hundred action classes. The average video length is three minutes. We follow Maaz et al. (2023) to use GPT-3.5 to evaluate the open-ended questions.

**IntentQA.** IntentQA (Li et al., 2023) is a VideoQA dataset with daily social activities. It contains three types of contexts, situational, contrastive, and commonsense contexts to provide context for intent understanding from videos.

**CLEVRER.** CLEVRER (Yi et al., 2019) is a collision event-based video dataset that studies the temporal and causal structures behind the videos of simple objects. It includes 20,000 synthetic videos of colliding objects and more than $300,000$ questions and answers. It has four types of questions: descriptive (for example, "what color"), explanatory ("what's responsible for"), predictive ("what will happen next"), and counterfactual ("what if"). We report per question the accuracy of the CLEVRER, and submit to the official evaluation server (Eva) to get results from the test set.

### 3.2 BASELINES

Our first baselines are the flagship VLM (GPT-4o family, Gemini 1.5 family, and open-sourced LLaVA-OneVision family.). We use 1 fps sampling to extract frames, a standard way to perform video question answering. We then use the questions, all possible answers, and the frames as prompts to the VLM to reproduce the results. The details can be found in Section 3.3.

VideoAgent (Wang et al., 2024a) is an agent-based system with iterative selection. From an initial state of captions of uniformly sampled video frames, VideoAgent iteratively 1) identifies if more information is needed, 2) retrieves a new video frame with high relevance to the query in step 1), and adds the frame caption to its state until there is sufficient information to answer the query.

VideoAgent (Fan et al., 2024) is another LLM-based multimodal tool-use agent. First, it represents videos as a structured memory with two components: a temporal memory that stores text descriptions of short video segments, and object memory tracking and storing occurrences of objects and persons. Then it answers queries by invoking tools for querying both temporal and object memory, retrieving video segments, and visual question answering.

MoReVQA (Min et al., 2024) is a VideoQA model that uses modularity and multistage planning. First, it takes the question and passes it through an event parsing LLM, then it uses a grounding LLM, then a reasoning LLM, then a prediction LLM to get a final answer. Both the grounding stage and the reasoning stage access the video. The main benefits of this approach are that it is easier to interpret the results, the planning/execution traces are more grounded, and there are improvements in accuracy compared to previous approaches that use a single stage.

IG-VLM (Kim et al., 2024), which stands for Image Grid Vision Language model, is a method for video question answering that uses a zero-shot approach with just a vision language model. First, the video is transformed into a series of images in a grid layout. Thus, this grid is one larger image. Thus, the input does not need to be a video, but can instead be one larger image. IG-VLM outperformed previous baselines on nine out of ten benchmarks that they used.

VideoTree (Wang et al., 2024b) builds tree-based representation of videos by recursively (1) clustering visual embeddings of video frames, (2) for each cluster captioning its keyframe and scoring the caption's relevance to the query (3) for relevant clusters repeating 1 and 2, adding the sub-clusters as children in the tree. Queries are answered by traversing the tree and concatenating keyframe captions as a textual description of the video for the LLM. We implemented VideoTree on the CLEVRER dataset, based on the official codebase.

LVNet (Park et al., 2024) is a keyframe selection framework with a Hierarchical Keyframe Selector module which is composed of three submodules: Temporal Scene Clustering (TSC), Coarse Keyframe Detector (CKD), and Fine Keyframe Detector (FKD). It initially begins by processing dense frames and keywords and progressively exploits heavier and more performance-oriented modules on a small set of frames to reduce the keyframe candidates.

## 3.3 IMPLEMENTATION DETAILS

In the following, we list descriptions of how we setup and run different VLM inferences on different datasets. As VAP is agnostic to VLM, we evaluated VAP on three VLMs: Gemini 1.5 family, GPT-4o family, and LLaVA-OneVision family. For all datasets except CLEVRER, we use $n = 32$ frames for VLM inference. For CLEVRER, we use $n = 5$ as each video is only 5-seconds long and we adopt the standard 1 fps. For IntentQA, we use $n = 12$ for fair comparison with baseline LVNet. We resize all videos to 720 x 480 resolution as CogVideoX only supports this format. More concretely, the initial 32 frames are encoded with the pre-trained 3D Causal VAE. Noise is added to the initial latents based on strength and the current timestep, and scaled according to the DPM scheduler's noise sigma. The prompt texts (containing all questions and possible answers), the noisy latents, the time step embeddings, and the rotary positional embeddings are the conditioning signals. At denoising, the pre-trained transformer layers from CogVideoX will predict the noise and the latents are updated iteratively based on inference steps. We use 50 steps of denoising and a scale factor of 1.15258426. No masking is used. Then, the updated latents from the diffusion model will be leveraged by RIFE to interpolate: the latents are chunked and for each chunk, a latent frame is interpolated with the guidance of SSIM, whose threshold we can adjust to reach the number of frames. After interpolation, we have all the latent frames. We next explain inference on each VLM.

The Gemini 1.5 models (Reid et al., 2024) are one of the strongest multimodal VLMs. For Gemini, we use the Google API which can take video as direct input for standard uniform sampling baselines. Security filtering may filter out some answers for video input due to mandatory setting, so the results may differ from the official report (Reid et al., 2024). After gathering the responses, we parse the answered choice, or leave the answer as is for yes/no or open-ended answers.

GPT-4o (OpenAI, 2024) is OpenAI's flagship VLM model for reasoning across vision and text. We used the OpenAI API which only takes in images not videos. We use OpenCV to extract frames from videos (1 fps for standard uniform sampling baselines). Following OpenAI recommendation, we convert the extracted frames to Base64 encode images for GPT-4 to process. The frames are resized per GPT-4 requirement. GPT-4 has non-adjustable safety settings that may filter out some answers,

Table 1: Comparison of test set accuracy on zero-shot video QA tasks on EgoSchema, NExT-QA, ActivityNet-QA and CLEVERER datasets (standard baselines results are reproduced). VAP achieves state-of-the-art results on EgoSchema, NExT-QA, CLEVERER, and achieves competitive results on ActivityNet-QA.

| Model | VLM | EgoSchema | | NExT-QA | | | | ANet-QA | Intent-QA | CLEVRER | | | | |
|---|---|---|---|---|---|---|---|---|---|---|---|---|---|---|
| | | Sub. | Full | Tem. | Cau. | Des. | Avg. | Test | Avg. | Des. | Exp. | Pre. | Cou. | Avg. |
| *Standard baselines, 1fps uniform sampling* | | | | | | | | | | | | | | |
| LLaVA-OneVision (repro.) | LLaVA-OV-7B | 62.4 | 60.3 | 76.0 | 79.5 | 85.5 | 79.4 | 56.5 | - | - | - | - | - | - |
| LLaVA-OneVision (repro.) | LLaVA-OV-72B | 64.2 | 62.5 | 77.8 | 81.2 | 86.1 | 80.9 | 61.9 | 66.2 | - | - | - | - | - |
| Gemini 1.5 Flash (repro.) | Gemini 1.5 Flash | 63.3 | 60.2 | 62.9 | 65.9 | 70.4 | 65.7 | 50.9 | - | 42.9 | 9.7 | 53.2 | 14.4 | 30.1 |
| Gemini 1.5 Pro (repro.) | Gemini 1.5 Pro | 67.5 | 65.8 | 66.7 | 71.8 | 73.2 | 70.4 | 52.2 | - | 47.9 | 16.3 | 45.1 | 10.6 | 30.0 |
| GPT-4o mini (repro.) | GPT-4o mini | 65.1 | 63.3 | 55.3 | 57.5 | 66.7 | 58.3 | 57.8 | - | 34.6 | 14.9 | 45.6 | 12.7 | 27.0 |
| GPT-4o (repro.) | GPT-4o | 69.2 | 67.8 | 67.6 | 71.9 | 75.8 | 71.2 | 58.8 | - | 38.2 | 28.3 | 65.0 | 24.1 | 38.9 |
| *Frame selection baselines* | | | | | | | | | | | | | | |
| VideoAgent (Wang et al., 2024a) | GPT-4 | 60.2 | 54.1 | 64.5 | 72.7 | 81.1 | 71.3 | - | - | - | - | - | - | - |
| VideoAgent (Fan et al., 2024) | GPT-4 | 62.8 | 60.2 | - | - | - | - | - | - | - | - | - | - | - |
| MoReVQA (Min et al., 2024) | PaLM-2 | - | 51.7 | 64.6 | 70.2 | - | 69.2 | 45.3 | - | - | - | - | - | - |
| IG-VLM (Kim et al., 2024) | GPT-4V | - | 59.8 | 63.6 | 69.8 | 74.7 | 68.6 | 58.4 | 64.2 | - | - | - | - | - |
| VideoTree (Wang et al., 2024b) | GPT-4 | 66.2 | 61.1 | 67.0 | 75.2 | 81.3 | 73.5 | - | 66.9 | 38.1 | 11.3 | 42.6 | 9.8 | 31.9 |
| LVNet (Park et al., 2024) | GPT-4o | - | 61.1 | 65.5 | 75.0 | 81.5 | 72.9 | - | 71.7 | - | - | - | - | - |
| VAP (Ours) with LLaVA-OV | LLaVA-OV-72B | 65.0 | 63.2 | **77.9** | **82.1** | **86.1** | **81.4** | **64.6** | - | - | - | - | - | - |
| VAP (Ours) with Gemini 1.5 | Gemini 1.5 Pro | 67.9 | 66.0 | 68.3 | 72.1 | 73.4 | 71.1 | 56.4 | - | 47.9 | 16.4 | 46.0 | 16.8 | **40.5** |
| VAP (Ours) with GPT-4 | GPT-4 | 67.5 | 66.7 | 68.8 | 76.1 | 72.5 | 73.2 | 59.8 | - | - | - | - | - | - |
| VAP (Ours) with GPT-4o | GPT-4o | **69.4** | **68.1** | 69.2 | 77.4 | 70.8 | 73.8 | 61.3 | **72.2** | 38.8 | **28.7** | 64.8 | **26.1** | 37.3 |

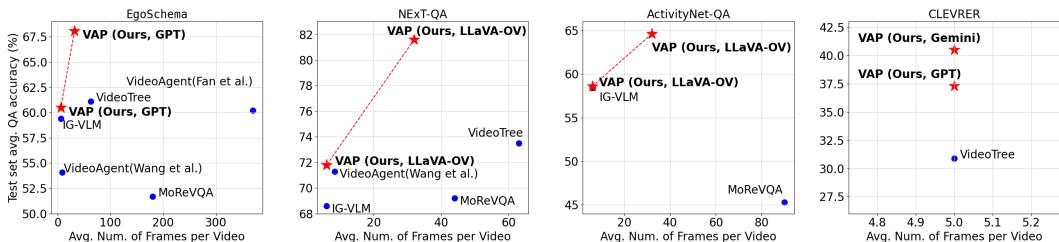

Figure 3: VAP also outperforms frame selection baselines with $n = 6$ and $n = 32$ frames per question, respectively, when compared to models with the same or a smaller number of selected frames.

so the results of GPT-4o may differ from the official blog (OpenAI, 2024). Similarly to the Gemini models, we parsed the generated response based on the answer type.

LLaVA-OneVision (LLaVA-OV) (Li et al., 2024a) is a recent open-sourced family of VLMs that achieved state-of-the-art results on single-image, multi-image and video tasks, with strong transfer learning performances. The results of LLaVA-OneVision are evaluated from the LMMS framework (Bo Li* & Liu, 2024), following the official guideline. We do not include LLaVA-OV results on CLEVRER because the only LMMS dataset that contain CLEVRER data is MVBench (Li et al., 2024b), but MVBench only contains a subset of CLEVRER, making it hard to compare with full-set CLEVRER results from other models.

## 3.4 RESULTS

**Main results.** We show results of VAP in Table 1. On EgoSchema, VAP achieves $69.4\%$ accuracy on the subset and $68.1$ on the entire test set, outperforming reproduced flagship VLM baselines, as well as previous work on frame selection for VLMs. On NExT-QA, VAP achieves higher accuracies than all baselines on all questions types: temporal (Temp.), causal (Cau.) and descriptive, with a state-of-the-art average accuracy of $81.4\%$. On ActivityNet-QA (ANet-QA), VAP achieves state-of-the-art VideoTree results of $64.6\%$. On CLEVRER, VAP achieves better accuracies on descriptive (Des.), explanatory (Exp.), counterfactual (Cou.), and state-of-the-art zero-shot accuracy with $40.5\%$. This shows that VAP is more effective in selecting key frames for QA tasks than standard VLMs or recent SOTA frame selection methods.

**Reasoning tasks.** In particular, VAP performs better on tasks requiring strong reasoning, i.e., temporal and causal questions in NExT-QA and explanatory and counterfactual questions CLEVRER. These questions focus on reasoning over the whole video dynamics and causes and consequences of person or object interactions, for example, why and how two people in the video interact in certain ways, what is responsible for an object collision, and what if certain objects had been removed. Answering these questions requires a model to select contextually important key frames of pivotal actions leading to the outcomes. VAP outperforms previous baselines, including standard VLMs and frame selection methods, and demonstrates its ability to select such consequential frames.

**Frame efficiency.** We showed the performance with respect to the number of frames used for the standard flagship VLMs in Figure 1 in introduction. VAP achieves better results using $n = 32$ frames in EgoSchema, NExT-QA, ActivityNet-QA than standard flagship VLMs with 180, 44 and 90 frames, providing 5.6×, 1.5×, and 2.8× frame efficiency improvements, respectively. On CLEVRER, VAP achieves better results using the same $n = 5$ frames with baselines. These results show that VAP greatly improves the efficiency of long-form video QA for the flagship VLM inference. Next, in Figure 3, we also show the performance compared to other frame selection baseline methods. Because some baseline methods select fewer than $n = 32$ frames, we also provide results with $n = 6$. VAP outperforms recent frame selection baselines when compared to those with the same or a smaller number of selected frames, suggesting that VAP is more proficient in selecting the most relevant frames compared to other baselines.

### 3.5 QUANTITATIVE ANALYZES

**Varying number of selected frames.** We are curious whether increasing the number of selected frames will increase performance. We conduct experiments on EgoSchema and ActivityNet-QA, with a number of frames ranging from $n = 6$ to $n = 48$, the maximum number of frames we can select due to generation capacity of the video generation model CogVideoX. Due to cost, we select the GPT-4o mini as the VLM for this comparison. We report the results in

Table 2: EgoSchema and ActivityNet results by different numbers of selected frames.

| # frames | Acc. (%) (w/ GPT-4o mini) | |
|---|---|---|
| | **EgoSchema** | **ANet-QA** |
| 6 | 54.2 | 50.2 |
| 16 | 58.6 | 53.3 |
| 32 | 63.3 | 57.8 |
| 48 | 63.5 | 57.7 |

Table 2. The performance on VAP with GPT-4o mini consistently improves when number of frames $n$ increases from 6 to 32, and plateaus when frame is greater than $n = 32$, therefore we report $n = 32$ results for other sections of the paper. Future work may explore using more frames for more capable models such as GPT-4o or Gemini 1.5. The results suggest that $n = 32$ is optimal for VAP and additional visual frames may be redundant or unnecessary for the questions.

Table 3: EgoSchema and ActivityNet results by different numbers of initial frames.

| # frames | Acc. (%) (w/ GPT-4o mini) | |
|---|---|---|
| | **EgoSchema** | **ANet-QA** |
| 6 | 38.0 | 34.1 |
| 16 | 51.7 | 53.3 |
| 32 | 63.3 | 57.8 |
| 64 | 63.8 | 58.4 |
| 90 | 63.2 | 57.2 |

**Varying number of initial frames.** Our next comparison focuses on varying the number of initial frames fed to the generation model to generate and interpolate all frames in the latent space. This is different from the last comparison, which focuses on the number of final selected frames for VLM inference. We conducted experiments on EgoSchema and ActivityNet-QA, with the number of frames ranging from $n = 6$ to $n = 90$, where $n = 90$ is the limit of sampling ActivityNet-QA frames at 1 fps. We also select GPT-4o mini as the VLM for this comparison. We report the results in Table 3. The performance on VAP with GPT-4o mini consistently improves when the number of frames $n$ increases from 6 to 32, stays very close from $n = 32$ to $n = 64$, and drops slightly at $n = 90$. The first increase may suggest that more initial frames may benefit the generation model in capturing the full context and dynamics of the video. However, as more frames are available, redundant frames can fully expose all task-relevant video dynamics and therefore make generation easy, and making it less likely for VAP to select pivotal frames that change scenes or actions and therefore are task-relevant. In this paper, we use $n = 32$ for all tasks except CLEVRER for ease of implementation and computation.

**Reasoning abilities compared to frame selection baselines.** We are also interested in the reasoning capabilities of other frame selection baselines, particularly for the CLEVRER data set. CLEVRER is different from other datasets as it comprises of simple rendered objects but includes challenging Explanatory (Exp.), Predictive (Pre.) and Counterfactual (Cou.) tasks. These can be particularly hard for previous frame selection models, as they rely on captioning models to extract visual information and use the captions instead of the frames to feed to the VLM for QA tasks. From Table 1, VAP have 154.0%, 71.43%, and 8% relative performance gains over VideoTree on explanatory, counterfactual, and predictive tasks, respectively, demonstrating the benefit of using visual frames instead of captions on challenging visual reasoning tasks.

### 3.6 QUALITATIVE ANALYZES

We provide qualitative analyzes on the frames selected by VAP on EgoSchema and CLEVRER, as illustrated in Figure 4. On EgoSchema (Figure 4a), the first example requires frames towards the end of the video to answer the question, and VAP correctly selects most frames towards the end. Presumed key frame (a person taking out a phone) from the initial frames is different from the rest of the scene but is question relevant, and VAP are able to extract frames adjacent to this key frame. In

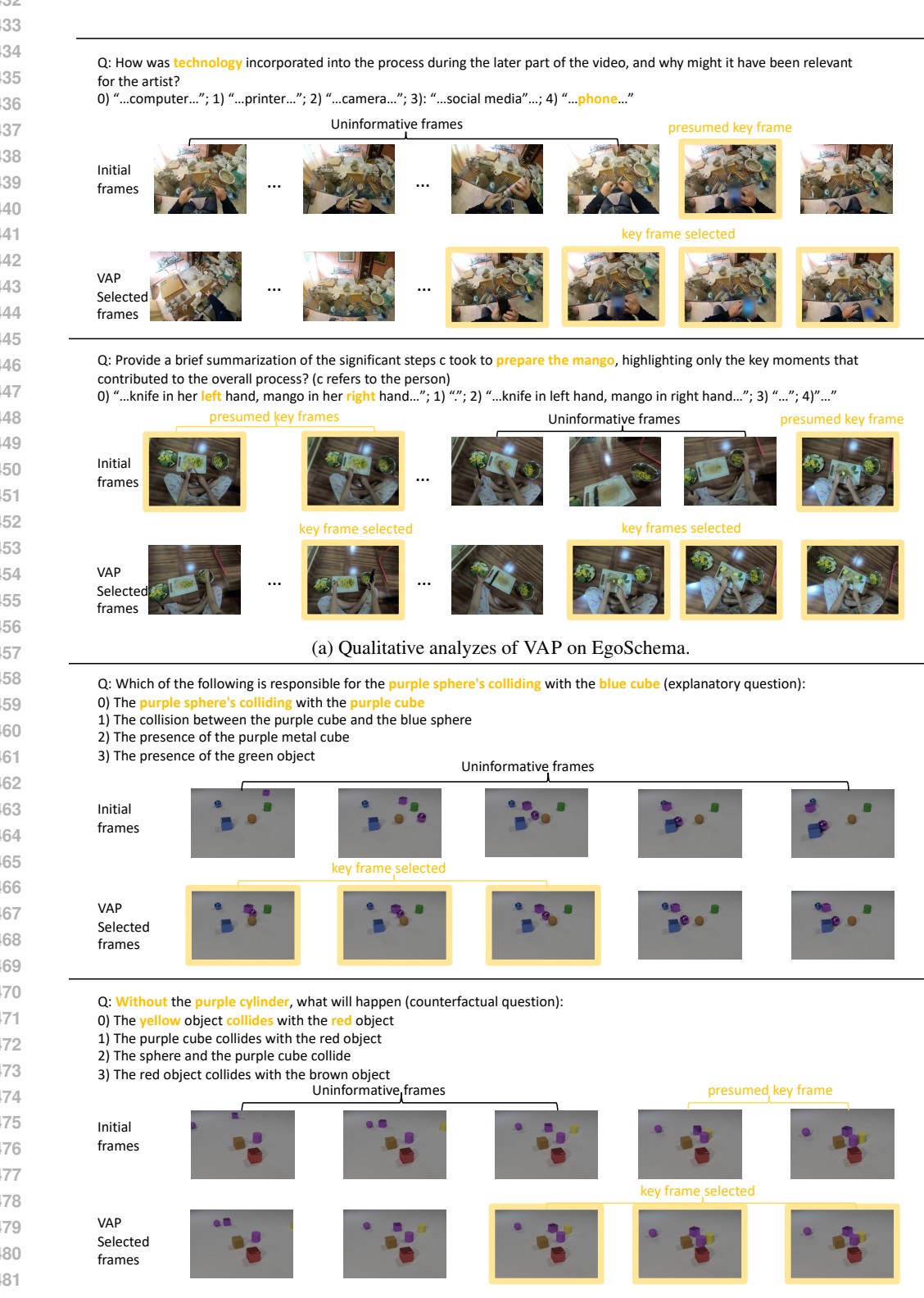

(a) Qualitative analyzes of VAP on EgoSchema.

(b) Qualitative analyzes of VAP on CLEVRER.

Figure 4: Qualitative analyzes show that VAP can select key transitioning frames relevant for question-answering, including action frames towards end of video (first example), key frames at both the beginning and at the end of the video (second example), unseen key collision frames (third example), and frames that demonstrate the pivotal pre-collision and post-collision moments for counterfactual question (fourth example).

the second example, the question can be answered by both the beginning and the end of the video, despite the visual detail differences. VAP is successfull at selecting the frames at both ends of the video, especially at the end of the video where the scenes are most useful for the answer. In both cases, VAP is successful in finding frames that contain crucial tasks-relevant frames, although they differ from other uninformative frames in visual details.

On CLEVRER, the explanatory and counterfactual questions are particularly hard because they require reasoning over positions and interactions over many objects. We show two examples for each type of question (Figure 4b). In the first CLEVRER example, a purple sphere first collides with a purple cube, and the purple sphere then collides with the blue cube. The question is to ask which object is responsible for the later collision. None of the initial frames actually shows the first collision, which is the key to the answer. VAP, however, can successfully pick up the key scenes of the first collision, as these frames are visually different from initial frames but are question-relevant. In the second example, a yellow object outside the scene comes in and collides with the purple cylinder. The counterfactual question is to ask what happens if there is no purple cylinder (the correct answer is that the yellow object will collide with the red object). Answering this question requires frames that show the trajectories of the yellow object, which are frames containing the pivotal pre-collision and post-collision moments. VAP successfully picked up these frames, demonstrating its effectiveness in selecting key frames of crucial transitioning trajectories.

## 4 RELATED WORK

**Active perception, active feature and input acquisition.** Active perception (Tenenbaum, 1971; Bajcsy, 1988; Aloimonos, 2013; Pulvermüller & Fadiga, 2010; Bajcsy et al., 2018; Satsangi et al., 2020; Zaky et al., 2020; Zhang & Fisac, 2021) refers to the theory in which an agent actively acquires its sensory input to optimize perception based on feedback from an a priori knowledge model. Similar ideas include active sensing (Ji & Carin, 2007; Yu et al., 2009; Yang et al., 2016; Yin et al., 2020), active feature acquisition (Saar-Tsechansky et al., 2009; Shim et al., 2018; Lewis et al., 2021), and active data acquisition (Kossen et al., 2022; Lai et al., 2023). In this work, we leverage the active perception framework by using a video generation model to represent a priori knowledge and selecting key frames as a data acquisition process for VLM inference.

**Frame selection methods for long-form video question-answering.** Long-form video question answering presents significant challenges for large vision-language models (VLMs) due to the computational cost. Recent efforts have aimed to reduce the number of input frames to enhance efficiency. VideoAgent (Wang et al., 2024a) employs an iterative frame selection method to identify a minimal but sufficient set of frames for prediction. Similarly, Fan et al. (2024) introduce a structured memory that stores event descriptions and object tracking states, to localize features when given an input query. MoReVQA (Min et al., 2024) uses modularity and multistage planning and allows interpretation. IG-VLM (Kim et al., 2024) uses a series of images in a grid layout as input to VLM VideoTree (Wang et al., 2024b) constructs a hierarchical tree structure to represent video information through multiple rounds of sampling and captioning. In contrast, our approach requires only a lightweight pre-trained video generation model, eliminating the need for captioning models or complex data structures. By selecting frames in a single round rather than multiple iterations, our method offers a simpler and more efficient solution. Using the principles of active perception, we focus on frames that diverge most from expectations from the video generation model.

## 5 CONCLUSIONS

In conclusion, we have presented Video Active Perception (VAP), a novel, training-free method that significantly improves both the efficiency and effectiveness of long-form video question answering. Drawing on active perception theory and employing a lightweight text-conditioned video generation model to represent prior world knowledge, VAP selects key frames that are most informative, those that diverge the most from expected dynamics. Our extensive experiments demonstrate that VAP not only achieves state-of-the-art zero-shot performance on datasets like EgoSchema, NExT-QA, ActivityNet-QA, and CLEVRER but also improves efficiency by up to 5.6× in frames per question. These results highlight the potential for integrating intelligent data acquisition strategies and prior knowledge into vision-language models. We believe that this approach opens new avenues for research in efficient multimodal reasoning and has significant implications for real-world applications that involve processing large volumes of video data by large Vision-Language Models.

## 6 REPRODUCIBILITY STATEMENT

The authors will provide an anonymous repository link in a comment to reviewers and area chairs when discussion forums open, according to ICLR policy. The code base will include evaluation scripts of both proprietary VLMs and open-sourced LLaVA-OV for the proposed VAP. The code has been anonymized and will not include any author information. The code will be made public upon acceptance on paper. Additional implementation details are available in the Appendix.

## 7 ETHICS STATEMENT AND LIMITATIONS

The proposed method performs inference on datasets collected from video on the Internet, which may reflect biases. The method also relies on existing proprietory or open-sourced vision-language models which may demonstrate a variety of security vulnerabilities, such as adversarial triggers to generate undesirable outputs and privacy risks such as memorization of training data.

This work relies on video generation models and large vision-language models, and changes in both models can directly impact the performance of the proposed work, limiting the reproducibility of the method. This work also does not include training or fine-tuning of either the generation or VLM models that could potentially improve the performance and efficiency of the proposed method. Future work could also explore selecting even more frames with VAP to improve performance.

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

# A   APPENDIX

## A.1   VAP IMPLEMENTATION DETAILS

We provide the main prompt for the video generation model in 4. We provide a detailed prompt with examples to leverage the video generation model's capacity as much as possible.

In terms of the video generation model, including the video diffusion model, 3D VAE, and interpolation model, we refer the details of the model checkpoints, training and inference processes to Yang et al. (2024).

## A.2   EXPERIMENTAL DETAILS (ACTIVITYNET-QA)

The ActivityNet-QA test set contains 8000 QA with open-ended answers. For reproducing baselines, while GPT models can hold temporal context, they do not support videos directly. Hence, frames were sampled at 1 fps and provided to the GPT model. The videos were provided directly to the Gemini models. The format for the prompt is provided in 5. Following standard evaluation (Achiam et al., 2023), we use GPT-3.5 to evaluate the open-ended answers.

## A.3   EXPERIMENTAL DETAILS (NEXT-QA)

We test the reasoning capabilities of the current state-of-the-art vision language models in a zero-shot setting. The test set contains about 8500 multi-choice QA with five canditate options. For reproducing baselines, while GPT models can hold temporal context, they do not support videos directly. Hence, frames were sampled at 1 fps and provided to the GPT model. The videos were provided directly to the Gemini models. The format for the prompt is provided in 6 for Gemini models and in 7 for GPT models. The prompts blocked or answered in an incorrect format (not outputting the option) by these models were dropped. The drop rate for each model is provided in 8.

## A.4   EXPERIMENTAL DETAILS (CLEVRER)

We evaluate on CLEVRER's test set, which contains 5,000 videos and 76,340 QA pairs. For multiple choice questions, we report both per-option accuracy and per-question accuracy. Per-option accuracy measures the overall correctness of selected options across all questions, and per-question accuracy measures the overall correctness of questions which require all choices to be selected correctly. For reproducing baselines, in both Gemini 1.5 and GPT-4o we sample the videos at 1 fps. We use the prompts in 10 and 11 for multiple choice and single word answer questions respectively. Furthermore, we report in 9 the proportion of multiple choice questions for which Gemini 1.5 and GPT-4o do not select any of the options, classifying them all as incorrect. For Gemini 1.5 Pro, we initially observed that no options were selected for 25.4% of the multiple choice questions, a significantly higher rate than Gemini 1.5 Flash and both GPT-4o variants. After re-evaluating Gemini 1.5 Pro on these questions, the rate of multiple choice questions with no selected options dropped to 14.2%.

## A.5   EXPERIMENTAL DETAILS (EGOSCHEMA)

We evaluated EgoSchema on the entire set of 5,000 question answer pairs. Each question was multiple choice, with 5 answers. We calculated the percentage of correct multiple choice answers on the entire set of 5,000 questions. Each video in EgoSchema is 3 minutes long, which is 180 seconds. For reproducing baselines, in order to sample the video frames, we processed one frame per second, and passed in the array of 180 frames. In terms of models, we evaluated EgoSchema on GPT-4, GPT-4o, and Gemini. The specific prompt format for each of these is shown in Fig. 12.

System: You are an advanced video generation model designed to predict plausible future video dynamics based on limited input. Your primary goal is to use your extensive prior knowledge of the world to generate latent representations of how the video is expected to unfold, given:
A few initial frames from the video;
A question about the video;
Possible answers to the question;
These generated dynamics will assist in identifying key frames in the actual video that are most informative for answering the question.

User: Your Task:
1) Analyze the Initial Frames:
1a) Examine the provided initial frames to understand the setting, context, characters, objects, and any ongoing actions or events.
1b)Extract visual cues that indicate the environment (e.g., indoor, outdoor, time of day) and participants (e.g., people, animals, objects).
2) Incorporate the Question and Possible Answers:
2a) Read the question carefully to determine what information is being sought.
2b) Consider each possible answer to understand different potential outcomes or scenarios.
2c) Use this information to guide your expectations of how the video might progress.
3) Generate Expected Video Dynamics:
3a) Using your prior knowledge and the initial frames, predict plausible sequences of events that align with the context and are relevant to the question.
3b) Focus on generating dynamics that would lead to scenarios described in the possible answers.
3c) Create latent representations that capture these expected continuations, including scenes, events, actions, and transitions.

Input Information:
1) Question about the video: {Question}
2) Possible Answers: {Answers}
3) Initial Frames: {Initial Frames}

Instructions:
1) Leverage Prior Knowledge:
1a) Utilize your understanding of real-world behaviors, cause-and-effect relationships, and typical sequences of events. 1b) Incorporate common sense and logical reasoning to predict what is likely to happen next.
2) Focus on Relevance:
2a) Ensure that the generated dynamics are directly relevant to the question and possible answers. 2b) Highlight events or actions that would help distinguish between the different answers.
3) Maintain Consistency: 3a) Keep the generated content consistent with the visual information in the initial frames (e.g., same characters, objects, setting). 3b) Avoid introducing improbable elements that contradict the initial context.

Example:
Initial Frames: Show a person standing at a crosswalk, waiting for the light to change. Question: "What does the person do after the light turns green?" Possible Answers: "They cross the street." "They turn around and walk away." "They start jogging along the sidewalk." Your Generated Dynamics Should:
Predict the likely actions following the initial frames, considering each possible answer. Generate latent representations where: The person crosses the street when the light turns green. The person changes their mind and walks away from the crosswalk. The person begins jogging along the sidewalk instead of crossing.

Output Format:
Provide latent representations (in your internal format) that correspond to the expected video dynamics. Ensure that these latents encapsulate the visual and temporal progression of events relevant to the question and answers.

Additional Notes:
Attention to Detail: Capture subtle cues from the initial frames that might influence the outcome (e.g., the person's expression, items they are carrying, environmental conditions). Diversity in Scenarios: While maintaining plausibility, consider multiple potential developments that are consistent with the possible answers. Purpose of Generation: Remember that the goal is to identify discrepancies between expected and actual video content to select informative frames for further analysis.

Table 4: GPT-4o prompt for VAP
.

Answer the following question about the video using only a word or two. Never say "unknown", "N/A" or "unsure", instead provide your most likely guess. Note that "where" questions refer to locations and not relative positions. Answer binary questions with yes or no.
Question: {Question} Answer:

Table 5: Gemini and GPT-4 prompt for ActivityNet

You are provided with a video followed by a question and choices. Answer the questions providing only the number of the correct choice.
{Video} {Question} 0. {Choice 0} 1. {Choice 1} 2. {Choice 2} 3. {Choice 3} 4. {Choice 4}

Table 6: Gemini prompt for Next-QA

These are frames from a video that I want to upload. Answer the questions providing only the number of the correct choice.
{Video} {Question} 0. {Choice 0} 1. {Choice 1} 2. {Choice 2} 3. {Choice 3} 4. {Choice 4}

Table 7: GPT-4o prompt for Next-QA

| LLM | Drop Rate (%) |
|---|---|
| GPT-4o mini | 0.7 |
| GPT-4o | 2.0 |
| Gemini 1.5 Flash | 4.2 |
| Gemini 1.5 Pro | 4.7 |

Table 8: The percentage of points dropped for each model during evaluation due to the model blocking prompts or not answering the multiple choice.

| Model | No Answer Rate |
|---|---|
| GPT-4o | 0.016 |
| GPT-4o-mini | 0.017 |
| Gemini 1.5 Pro | 0.142 |
| Gemini 1.5 Flash | 0.032 |

Table 9: Proportion of CLEVRER multiple choice questions where no options were selected.

You will be provided frames from a video, sampled evenly across the video. You will also be given a question about the video and an enumerated list of options. Select all options that are correct. After explaining your reasoning, output your final answer in the format "Final Answer: comma separated list of correct option numbers". At least one option is correct, so always pick the option(s) that are most likely to be correct even if no option seems entirely correct.

{{video}}
Question: {{ question }}
Options:
{% for option in options %}
({{ loop.index0 }}) {{ option }}
{% endfor %}

Table 10: Prompt for CLEVRER multiple choice questions.

You will be provided frames from a video, sampled evenly across the video. Answer the question about the video using only a word or number. Never say "unknown", "N/A" or "unsure", instead provide your most likely guess. Answer binary questions with yes or no.

{{video}}
Question: {{question}} Answer:

Table 11: Prompt for CLEVRER binary questions.

> You will be given a question about a video and five possible answer options, where C refers to the person wearing the camera. You will be provided frames from the video, sampled evenly across the video.
> {video}
> Question: {question}
> Possible answer choices:
> (0) {option0}
> (1) {option1}
> (2) {option2}
> (3) {option3}
> (4) {option4}
> After explaining your reasoning, output the final answer in the format "Final Answer: (X)", where X is the correct digit choice. Never say "unknown" or "unsure", or "None", instead provide your most likely guess.

Table 12: Prompt for the EgoSchema dataset.

| Model | VLM | Frames per video | Perception score |
|---|---|---|---|
| GPT-4o mini | GPT-4o mini | 32 | 38.4 |
| GPT-4o | GPT-4o | 32 | 54.1 |
| VAP (ours) | GPT-4o mini | 32 | 40.6 |
| VAP (ours) | GPT-4o | 32 | **55.7** |

Table 13: VideoMME results. VAP demonstrates better results than baseline GPT, suggesting its effectiveness in choosing informative frames over very long videos.

## A.6 EFFICIENCY COMPARISONS.

Below we provide the peak memory, FLOPS, and total runtime of baseline methods and VAP (32 frames for GPT-4o and Gemini 1.5 Pro baselines, or frames determined by the implementation for other baselines, running on A100 machines).

We include the proportional tree-building time of VideoTree. The high FLOPS of VideoAgent is because it runs open-source VLMs for frame selection. The VAP overhead includes the causal 3D VAE encoding and decoding, the conditional diffusion model and the interpolation model.

From Table 14, VAP has the advantage of total runtime over VideoAgent and VideoTree, albeit with larger peak memory. It also saves significant FLOPS compared to VideoAgent, because the diffusion model is lightweight and the diffusion steps are moderate (50). This suggests that VAP has practical advantages in efficiency including the number of frames and total runtime.

## A.7 VIDEOMME RESULTS.

We also show results on very long videos. We choose VideoMME (Fu et al., 2024), which is a video benchmark for VLMs with diverse video types, multiple video durations, and breadth in modalities. We choose the long split, which contains the longest 300 videos in the dataset, with the average length being 44 minutes, ranging from 30-60 minutes. We use GPT-4o mini and GPT-4o based on the LLMS-eval codebase.

We include the results in Table 13. VAP demonstrates better results than baseline GPT-4o, suggesting its effectiveness in choosing informative frames over very long video dynamics. We believe as the video generation model advances, our method can be empowered by future work with better long-term dependency reasoning capacity.

| Model | Peak Memory | Average FLOPS | Total runtime (seconds) |
|---|---|---|---|
| GPT-4o | N/A (NO GPU) | N/A (NO GPU) | 187 |
| Gemini-1.5 Pro | N/A (NO GPU) | N/A (NO GPU) | 236 |
| VideoAgent (Fan et al., 2024) | 46GB | 8401 GFLOPS | 495 |
| VideoTree (Wang et al., 2024b) | 23GB | N/A (small GPU usage) | 378 |
| VAP + GPT-4o | 54 GB | 87 GFLOPS | 246 |
| VAP + Gemini-1.5 Pro | 54GB | 87 GFLOPS | 297 |

Table 14: Efficiency analyses based on peak memory, FLOPS, and total runtime. The results above suggest that VAP has practical advantages in efficiency such as total runtime.

