# OpenReview forum: "Video Active Perception: Efficient Inference-Time Long-Form Video Understanding with Vision-Language Models"
_ICLR.cc/2025/Conference — Submitted to ICLR 2025_

### Official Review · Reviewer_hwaL · 2024-10-22

**Soundness:** 3
**Presentation:** 3
**Contribution:** 3
**Rating:** 6
**Confidence:** 5

**Summary:**

VAP introduces a novel method for keyframe selection using 3D VAE's and diffusion models. By comparing similarity (in the latent space) of uniformly sampled frames to the densely encoded frames, VAP selects visually different keyframes which can then be used with frontier LMMs, improving their performance, and achieving SOTA performance on multiple benchmarks.

**Strengths:**

* Interesting/novel approach to selected key frames (video generation model/diffusion).
* SOTA performance on several benchmarks
* Show that video diffusion models can be used to 'predict' video frames

**Weaknesses:**

* In Figure 3, Avg. number of frames are compared instead of FLOPs/latency. using diffusion models may make the approach more computationally expensive/higher latency compared to existing methods, while sampling less frames.
* Can you add evaluations on Video-MME/MVLU?

**Questions:**

Please see weaknesses.

---

### Official Review · Reviewer_bXhP · 2024-11-03

**Soundness:** 3
**Presentation:** 3
**Contribution:** 3
**Rating:** 5
**Confidence:** 5

**Summary:**

The paper introduces Video Active Perception (VAP), a method aimed at efficient inference-time understanding of long-form videos using vision-language models. VAP is designed to actively select keyframes during inference to reduce computational costs while maintaining performance.

**Strengths:**

VAP enhances long-form video QA efficiency by selecting key frames that diverge from a lightweight video generation model, achieving up to 5.6× efficiency improvement and outperforming other models on multiple datasets, demonstrating effective use of prior knowledge for better performance.

**Weaknesses:**

LVNet [1] introduced the hierarchical keyframe selection strategy for the very long-form video and achieves 61.1% accuracy using 12 frames with GPT-4o on the EgoSchema dataset. In contrast, Table 2 shows that VAP attains 58.6% accuracy with 16 frames using GPT-4o mini—a lower accuracy despite utilizing more frames. This discrepancy raises concerns about why VAP underperforms compared to LVNet on EgoSchema, even when processing more frames. I recommend that the authors provide an explanation for VAP's lower performance relative to LVNet. Additionally, comparing the performance on the NExT-QA dataset using approximately 12 frames would offer valuable insights. If VAP performs worse than LVNet when utilizing around 12 frames, it might suggest that the keyframe selection quality of VAP is less effective than that of LVNet. It would be beneficial for the authors to experiment with VAP using 12 frames and GPT-4o to directly compare its performance with LVNet. It would be the best if the author can show the performance of VAP on IntentQA and compare it to LVNet to prove its generalizability on diverse very long-form datasets.

Moreover, to ensure a fair comparison, it would be appropriate to adopt GPT-4 instead of GPT-4o when evaluating VAP against VideoTree. Otherwise, it remains unclear whether any performance boost is due to the VAP architecture or the use of a more powerful language model.

Another concern is the lack of analysis regarding computational cost and inference speed. Since VAP aims to be efficient at inference time, addressing this concern can be beneficial. Including metrics such as the number of inserted captions and inference speed in Table 1 would allow for a direct comparison of both accuracy and complexity among models.

[1] Park, Jongwoo, et al. "Too Many Frames, Not All Useful: Efficient Strategies for Long-Form Video QA." arXiv preprint arXiv:2406.09396 (2024).
[2] Li, Jiapeng, et al. "IntentQA: Context-Aware Video Intent Reasoning." Proceedings of the IEEE/CVF International Conference on Computer Vision. 2023.

**Questions:**

I have addressed this in the weakness section

---

### Official Review · Reviewer_sf66 · 2024-11-04

**Soundness:** 2
**Presentation:** 3
**Contribution:** 2
**Rating:** 6
**Confidence:** 4

**Summary:**

This paper introduces a frame selection method formulated as active perception, to efficiently handle long-form video understanding based on a video diffusion prior and a VLM. It is motivated to identify out-of-distribution (or, surprising) frames in a given video, using the diffusion prior as world knowledge (i.e., expected behavior). The authors first uniformly sample a small number of frames, and feed it through a diffusion model to interpolate and generate intermediate frame latents, also conditioned on question/answer text. Next, they compare the generated frames and the original frames in the diffusion latent space, identifying the most different frames as keyframes, subsequently using these for long-form video QA. The proposed method is validated on multiple benchmarks (eg: EgoSchema, NExT-QA, ActivityNet-QA, CLEVRER) based on various VLMs (GPT-4o, Gemini 1.5 Pro, and LLaVA-OV).

**Strengths:**

- The proposed method is very interesting and refreshing, as it relies on a video-diffusion prior to encode world-knowledge in a multi-modal LLM setup.
- The idea of keyframe sampling is extremely useful in long-form video understanding to mitigate associated costs.
- The authors have conducted extensive experimentation on mutliple long-video VQA benchmarks, making use of both open-sorce and proprietary LLMs.
- The paper is generally well-written and easy-to-follow.

**Weaknesses:**

- This paper mainly advertises the efficiency of the proposed pipeline as it relies of fewer frames. However, it also relies on many other components to sample such keyframes (eg: diffusion prior). It would be interesting to see whether it would translate to an efficiency gain that matters (eg: runtime, flops/memory of overall pipeline). Is there another way to justify efficiency measurement only in-terms of the number of frames (eg: significant reduction in prompting cost)?
- I have concerns about the generated latent frames. (1) A lot of details about the diffusion based generation and interpolation is missing (eg: which frames are generated: intermediate/subsequent, if any masking is used, exact nature of conditioning, number of denoising steps, model scale, what are the transformer layers in L146: are they already pretrained?). (2) I doubt the fidelity/usefulness of the generated frames. Can the model generate high-fidelity missing frames with reasonable motion that we may see in a 3min video? This can be easily evaluated by conveting the latent back to the pixel space and visualizing. It would be interesting to see how different they are from actual frames.
- In this method, initial frames are unoformly-sampled and the highest-different frames from these are fed to the VLM for video VQA. However, for questions that require normal/vanilla dynamics (eg: asking about the generic/prominent activity in a given video), should not have a better acuracy by sampling deviating (or, surprising) frames from the norm, right? This should be explained further, as otherwise the motivation of this work does not generalize.
- The performance gain seems to be minimal in most of the comparisons in Table 1. Also the comparison with other frame selection methods is not entirely fair (eg: others using GPT-4 and the proposed method using GPT-4o). Can the authors make a fair comparison, by running a few experiments with GPT-4?

**[Update]: Since the authors have addressed most of my concerns, I am raising my original rating from 3 to 6. There is still a confusion about efficiency (*e.g.* wall-clock time), yet I don't believe it warrents a reject rating--- as this work can be useful to the community. I suggest the authors encorporate all the discussions in the final version of the paper for improved clarity and better foundation of claims.**

**Questions:**

- How best can the authors justify efficiency measurements only w.r.t. the number of frames, and avoiding other metrics such as latency, computations or memory of the overall pipeline?
- Can the authors clarify missing details about how they exactly use the diffusion prior (eg: which frames are generated: intermediate/subsequent, if any masking is used, exact nature of conditioning, number of denoising steps, model scale, what are the transformer layers in L146: are they already pretrained?)?
- Can the authors visualize generated frames in pixel-space, and compare with real frames?
- How does the proposed method address questions about generic scenarios, as it always try to sample frames that are very different from the norm?

---

### Official Review · Reviewer_adHH · 2024-11-04

**Soundness:** 3
**Presentation:** 3
**Contribution:** 3
**Rating:** 6
**Confidence:** 4

**Summary:**

This paper proposes Video Activate Perception (VAP), a training-free method that employs a text-conditioned video generation model to represent prior world knowledge. VAP selects key frames that diverge the most from expected dynamics. An experiment shows that  VAP achieves SOTA zero-shot performance on EgoSchema, NExT-QA, ActivityNet-QA, and CLEVRER and also improves efficiency by up to 5.6x in frames per question.

**Strengths:**

- This paper introduces a novel approach, Video Activate Perception (VAP), which identifies key frames by contrasting generated video frames from a video synthesis model with frames from actual footage. This method offers a highly insightful strategy for key frames selection.
- This paper is well-written and easy to follow.

**Weaknesses:**

- To my understanding, the method first involves randomly sampling frames from the real video, which are subsequently excluded from being selected as key frames. If any of these randomly sampled frames contain critical information, it appears that the VAP approach might risk losing important content.
- The current work lacks exploration of comparative datasets for long video compression. To better validate its effectiveness, VAP should be evaluated on additional long video benchmarks like VideoMME, MLVU, and LongVideoBench.

**Questions:**

I noticed that the performance scores on the NeXT-QA dataset reported in the VideoTree paper are higher than those presented in this paper. Could there be an explanation for this difference？

---

### Official Review · Reviewer_DMMX · 2024-11-04

**Soundness:** 3
**Presentation:** 3
**Contribution:** 3
**Rating:** 6
**Confidence:** 5

**Summary:**

This paper introduces Video Active Perception (VAP), a method to improve the efficiency and performance of vision-language models (VLMs) in long-form video question answering (QA). VAP addresses the high computational costs of analyzing extensive video content by leveraging active perception theory. It uses a lightweight, text-conditioned video generation model to predict video dynamics and selects key frames that deviate from these predictions, optimizing input for inference.
This training-free, VLM-agnostic approach outperforms existing methods on datasets like EgoSchema, NExT-QA, ActivityNet-QA, and CLEVRER, achieving up to 5.6× efficiency improvements. VAP excels in tasks requiring complex visual reasoning, offering a simple and effective solution without needing captioning models or multi-round strategies.

**Strengths:**

- The introduction of Video Active Perception (VAP) is an innovative approach inspired by active perception theory. Unlike traditional methods, VAP leverages a lightweight, text-conditioned video generation model for efficient frame selection.
- The methodology is well-founded and clearly detailed, demonstrating rigorous experimental validation across multiple challenging datasets.
- The paper is well-structured and written clearly. The introduction effectively lays out the problem and the motivation for VAP, and the technical details are presented systematically.

**Weaknesses:**

- While the paper emphasizes efficiency gains in VLM inference by selecting key frames, it introduces substantial computational overhead through the use of additional models, including a 3D Causal VAE, a video diffusion model, and an interpolation model. The paper does not provide a detailed analysis or comparison of the total computational cost incurred versus saved, which raises questions about whether the claimed efficiency gains are realized in practical scenarios.
- The comparison with previous methods, such as VideoAgent using GPT-4, may not be entirely fair, as VAP uses a different version, GPT-4o, which could have different performance characteristics
- The performance improvements reported in Table 1, particularly on the EgoSchema dataset, are relatively modest (only a 0.2% increase over the GPT-4o baseline) and even underperform on CLEVRER in some cases. This raises concerns about the practical impact of the method, especially when the gains are not consistently significant across all datasets.
- The reliance on the video generation model will limit the use case when the video is extremely long. It's hard to capture the long-term temporal dependency using the current video generation model.

**Questions:**

- Could you provide a more detailed breakdown of the computational cost of VAP, including the overhead introduced by the 3D Causal VAE, video diffusion model, and interpolation model? How does the total computational cost compare to the savings from reduced VLM inference, and under what conditions would VAP be most beneficial? It would be helpful to see a discussion or empirical comparison quantifying the overall efficiency trade-offs.
- The comparison with VideoAgent and other methods using different versions of GPT models (e.g., GPT-4 vs. GPT-4o) introduces potential discrepancies. Can you discuss the impact of these differences on the results?
- The paper shows underperformance on CLEVRER compared to some baselines. Could you elaborate on the factors that contribute to these results? Are there specific aspects of CLEVRER (e.g., counterfactual or explanatory tasks) that make VAP less effective?
- Given the reliance on complex generative models, how well does VAP scale to real-world applications involving very long videos, such as continuous surveillance footage?

---

> ### Comment · Reviewer_DMMX · 2024-11-28
> **Response to Authors**
>
> Thank you for your efforts in providing additional experimental results for the rebuttal. After reviewing your response, my concerns have been partially addressed. However, I still have reservations about the computational cost and the practical value of the approach. I think the idea of leveraging a generative model to select important frames is interesting, in light of this, I am raising my score from 5 to 6.

---

### Meta-Review · Area_Chair_JhEk · 2024-12-24

**Metareview:**

The submission addresses the problem of inference time efficiency for (long-form) video understanding by selecting a small subset of frames conditioned on text prompts. The key intuition of the proposed approach is to leverage the prior "knowledge" encoded by a diffusion video generative model, and select the "keyframes" as those whose latent representations are the most distinct from the predicted diffusion latents. Evaluations are performed on EgoSchema, ActivityNet-QA, CLEVRER, etc. The submission received mixed ratings after rebuttal, including four borderline accepts (6) and one borderline reject (5). Below I summarize the main strengths and limitations of the submission, according to the reviews (after rebuttal discussion) and my own reading of the submission:

*Strengths:*
- All reviewers (and the AC) acknowledge that the proposed frame selection approach is novel and "refreshing". The method is also clearly presented.
- The empirical performance across multiple long-form video understanding benchmarks is strong.

*Weaknesses:*
- As the goal of keyframe selection is to efficiently process long-form videos, one concern shared by multiple reviewers (DMMX, sf66, bXhP, hwaL) is the computation overhead introduced by the proposed frame selection module.

Specifically, after rebuttal, reviewer hwaL had reservations on the latency of the proposed module (they suggested "cannot recommend this paper for publication" but the final rating remains "borderline accept"), reviewer sf66 had concerns on if the latency was calculated in a comparable manner for the proposed approach and the other baselines (e.g. GPT-4o API calls), they suggested "unable to recommend this paper for acceptance" with the final rating of "borderline accept". A similar concern was shared by DMMX in their last response to the authors. The AC shares this concern and believes the submission should be revised to clarify the actual inference-time efficiency benefits of the proposed approach before it is published. Therefore the AC cannot recommend the paper to be accepted by ICLR 2025.

**Additional Comments On Reviewer Discussion:**

The authors did a solid job in addressing the following questions:
- Performance comparison when using the same VLMs
- The robustness of the method against randomly selected initial frames
- The performance on additional benchmarks
- The quality of generated latent frames
- Comparison with additional baseline LVNet

As discussed above, the main remaining point to be addressed after rebuttal is to properly evaluate and present the inference-time efficiency benefits of the proposed approach. The AC also personally finds the argument on saving monetary costs to introduce additional confusion on inference-time effieciency. Overall, the manuscript should be revised to clarify these points before the submission could be published.

---

### Decision · Program_Chairs · 2025-01-22

Reject